# Food taboo practices among pregnant women in Deder town, Eastern Ethiopia, 2024

Abdi Tofik[1], Tesfaye Gobena[2], Addis Eyeberu[3], Adera Debella[3], Berhe Gebremichael[2], Mulugeta Gamachu[4], Alemayehu Deressa[2], Galana Mamo Ayana[2], Abdi Birhanu[4], Hamdi Fekredin Zakaria[2], Usmael Jibro[3]*, Ibsa Mussa[2]

1 Deder general hospital, Eastern Hararge zone, Oromia region, Addis Ababa, Ethiopia, 2 School of Public Health, College of Health and Medical Sciences, Haramaya University, Harar, Ethiopia, 3 School of Nursing and Midwifery, College of Health and Medical Sciences, Haramaya University, Harar, Ethiopia, 4 School of Medicine, College of Health and Medical Sciences, Haramaya University, Harar, Ethiopia

* usmiye20@gmail.com

## Abstract

### Background

Maternal nutrition during pregnancy is influenced by food taboo practices, which vary across cultural contexts. Food taboos during pregnancy significantly affect fetal outcomes by impacting maternal nutrition. Understanding these practices in Eastern Ethiopia is crucial for designing culturally appropriate interventions. This could contribute to a better understanding of food taboos practices and inform the development of culturally sensitive interventions to promote optimal nutrition during pregnancy. Therefore, the study aimed to assess the extent of food taboo practices among pregnant women in Deder town, Eastern Ethiopia.

### Method

An institutional-based cross-sectional study was conducted among 418 pregnant women. The study participants were selected by systematic random sampling. The data were collected using a structured interviewer-administered questionnaire. Data were entered into Epi data version 3.02 and then exported to SPSS version 25 for analysis. Binary logistic regression was fitted to identify factors associated with food taboo practices. P-value < 0.05 was used as a cut-off point for statistical significance.

### Results

The study showed that 56% (95% CI: 51.2, 60.8%) of pregnant women practiced food taboos. Pregnant women who were unable to read and write (AOR=3.36, 95%CI: 1.24, 9.16), did not have antenatal care (ANC) follow-up (AOR=2.04, 95%CI: 1.27, 3.29), food aversion (AOR=2.04, 95%CI: 1.31, 3.18), no additional meal practice (AOR=1.77, 95%CI: 1.14, 2.76), poor knowledge level (AOR=1.96, 95%CI: 1.24, 3.09), and unfavorable attitude (AOR=1.91, 95%CI: 1.22, 2.99) were significantly associated with food taboos practice.

**Data availability statement:** All relevant data are within the paper and its Supporting Information files.

**Funding:** The author(s) received no specific funding for this work.

**Competing interests:** The authors have declared that no competing interests exist.

## Conclusion

More than half of pregnant women practiced food taboos, indicating a significant public health concern. Culturally sensitive nutrition education and awareness programs at health facilities are necessary to address these practices and improve maternal nutrition outcomes.

## Introduction

Food taboos refer to dietary restrictions influenced by religious, cultural, or health beliefs [1]. Food taboos during pregnancy can have a significant impact on fetal outcomes. These taboos often result in the avoidance of certain nutritious foods, which can lead to maternal malnutrition [2–4]. Numerous studies from Asia [5–8] and Africa [8–12] have shown that women are compelled by traditional beliefs to forgo healthful diets throughout pregnancy and the postpartum period. The more prevalent taboos, albeit being of varied kinds, were related to the concurrent use of milk and milk products, eggs, linseed, fatty meats, fruits including mango, orange, pineapple, nuts, and vegetables [13–16]. Evidence showed that these food taboos contribute to the undernutrition of mothers and fetuses during pregnancy, which in turn harms their health [17]. In contrast to this, research done on the adaptive explanation of food taboo practice in Fiji showed that food taboos reduce women's chance of fish poisoning by 30% during pregnancy [18]. Another study done in India showed that pregnant women avoided food such as fruits to protect their fetuses which is acquired through learning [19].

Food taboo practices have been reported globally, with varying prevalence across regions. Globally, food taboo practices are prevalent, with rates reported at 70.2% in Malaysia [20], 37% in South Africa [16], and 66% in Nigeria [21]. In Ethiopia, reported prevalence ranges from 11.5% to 55.3% [10,12,13,22], with a systematic review estimating an average of 34.22% [14]. Regarding pregnant women's perception of food taboos, many women in Ethiopia have traditional beliefs relating to pregnancy and misconceptions about weight gain during pregnancy [10,23]

The avoidance of certain foods during pregnancy is thought to be influenced by a variety of variables, including culture, lack of nutritional counseling during antenatal care (ANC) visits, younger age, lower educational attainment, poor income, being multiparous, living in a rural area, and disliking of particular foods [12,14,24,25]. Nevertheless, a significant proportion of pregnant women refrain from eating particular foods because of cultural traditions or expectations. The concerns mentioned include worries about having a big baby, unpleasant vaginal discharge during birth, and skin conditions on the body [26]. Evidence shows that most women learn food taboos from their families and transmit them to society. Cultural traditions strongly influence dietary practices during pregnancy, as women often inherit food taboos through familial and societal transmission [10,16]. Evidence recommended that culturally appropriate interventions and indigenous knowledge about the food suggestions could be a good solution to enhance maternal nutrition [16,27]

The practice of food taboos has a significant negative influence on the health of new mothers and their babies. Dietary deficiencies, which have detrimental effects on both the mother and the newborn, are the most frequent consequence of food taboo [28]. The risk of premature birth, LBW, and less optimum growth and development of children may increase due to the mother's poor nutritional status during pregnancy [24,27,29]. Maternal death, psychological distress caused by poor fetal outcomes, and anemia are further serious effects of dietary

inadequacy for women [30]. Food taboos might also be adaptive and not always harmful to health [18,19,31].

Ethiopia is attempting to enhance the nutritional status of pregnant women by implementing policies and initiatives from international health recommendations. Despite interventions such as nutritional counseling and micronutrient supplementation [32–37], gaps remain in addressing culturally driven food taboos.

While Ethiopia has made strides in improving maternal nutrition and eating behaviors [38, 39], food avoidance during pregnancy remains a significant concern, particularly in Eastern Ethiopia, where cultural contexts differ as there is diverse food consumption behaviors among peoples live in eastern Ethiopia [40]. Understanding these practices is crucial to designing targeted, culturally sensitive interventions. This study aims to assess the extent and determinants of food taboos among pregnant women in Eastern Ethiopia.

## Materials and Methods

### Study design, area, and period

During the first to the last week of February 2022, a cross-sectional study was carried out in Deder Town's public health facilities in eastern Ethiopia. Deder, a town in Oromia, Ethiopia, is situated in the East Hararge Zone at a height of 2,117 meters (6,946 feet) above sea level. It is situated 458 kilometers east of Addis Ababa, the capital of Ethiopia. According to the population forecast for 2030, the town has a total anticipated population of 32,656, of which 16,987 were men and 15,669 were women. The community is home to 10 private clinics, five health posts, one general hospital, and one health center. More than 1.5 million patients from six districts are referred to Deder General Hospital. Deder General Hospital, established in 1934 by Non_Govermental Organization (NGO) and later shifted to a government hospital.

### Populations and criteria

Pregnant women who attended ANC follow-ups in public health facilities (Deder Hospital and health centers) were the study population. The study included all pregnant women attending ANC follow-ups at Deder Hospital and health centers, except those unable to respond during the data collection period.

### Sample size determination and sampling procedure

The minimum required sample size for the study was determined by using a single population proportion formula under the following statistical assumption: P = proportion of food taboo is 55.3% [41], (Z α/2 = Z score of 95% CI, d= Margin of error (5%).

$$n = \frac{(Z - /2)2 \times p(1-p)}{(d)2} = (1.96)^2 * 0.553 * 0.447 / (0.05)^2 = 380 .$$ Then after adding 10%

contingency rate for the non-response, the minimum sample size calculated was 418.

Two public healthcare facilities that offer ANC services (Deder Hospital and Deder Town Health Center) were chosen for the study. The determined sample size, n=418, was proportionally distributed taking into account the monthly patient flow rate of pregnant women for ANC utilization in the health institutions. As a result, the current study's determined sample size (418) included a total of 247 participants from Deder General Hospital and 170 participants from Deder Town Health Center. Systematic random sampling was used to select participants using ANC identification numbers, with the first sample chosen randomly between the first and second attendees.

## Data Collection procedure and quality control

The structured questionnaire was adapted from previous similar studies [10,15,42,43]. The tool had five parts which included sociodemographic characteristics such as age, residency, ethnicity, religion, educational status, occupational status, and monthly income; obstetrics-related characteristics such as parity, ANC follow-up, gestational age, and source of nutritional counseling; dietary pattern related items such as mean eating frequency, additional meal practice, and skipping meal practice; food aversion and craving related items (both open-ended and close-ended items) such as food aversion, type of food averted, reason for aversion, food craving, and reason for food craving; knowledge related questions; and attitude related question (both open-ended and close-ended items). The questionnaires were prepared in English versions and then translated into the local languages of the study area (Afan Oromo and Amharic). The questions were then translated back into English to ensure consistency and correctness of the translation. Any discrepancies in meaning between the original and the back-translated version are then fixed. The questionnaire was pre-tested on 5% of the sample size in the nearby Woreda health center before the real data collection, and adjustments were made depending on the results. Subsequently, the instrument's validity and reliability were examined. The reliability of the questionnaire was assessed using Cronbach's alpha, yielding a value of 0.83, indicating good internal consistency. Data were collected by seven trained health professionals, including five BSc midwifery and two BSc health officer. Data collection began on February 1, 2022, and ended on February 28, 2022. For data collectors and supervisors, a two-day training on the study's goal and data gathering instrument was provided. The supervisors and investigator double-checked the surveys every day. Each completed questionnaire was reviewed for consistency and completeness.

## Measurements

Pregnant women's attitudes regarding food taboo practices during pregnancy were assessed using seven 5-point Likert scale attitude questions. All of the questions were rated on a scale of 1–5 (1 being strongly disagreed, 2 disagreed, 3 neutral, 4 agreeing, and 5 strongly agreeing). The mean score was used as a cut point after the normalcy was checked, and participants with a mean score of 50% or higher were deemed to have positive attitudes toward food taboo practices, while those with a score below 50% were deemed to have negative attitudes [42,44]. To evaluate women's level of knowledge on food taboos during pregnancy, ten knowledge assessment questions were used. Each correct response received a score of 1, while incorrect answers received a score of 0. Following a normalcy check, a cut point of the mean score was determined. Participants were classified as having good knowledge if their mean score was 50% or higher, and as having poor knowledge if their mean score was less than 50% [42,44].

Additional meal practices: The Essential Nutritional Action (ENA 2015) recommends pregnant women eat at least one extra meal per day [45].

Frequency of regular meals: The Institute of Medicine recommends that pregnant women eat regular meals at least three times a day [46].

Food taboo: Food taboo is the deliberate avoidance of at least one food item for religious, cultural, and social reasons other than a simple dislike of food preference.

## Data processing and analysis

Epidata 3.02 was used to enter the data, which was then exported to SPSS 25.0 for additional statistical analysis. The characteristics of the study participants were described using means, medians, frequencies, and percentages. The significant variables associated with the outcome

variables were found using binary logistic regression analysis. Before fitting the model, the variance inflation factor (VIF) and tolerance test were used to evaluate the multicollinearity assumption for each covariate. All independent variables were fitted with the bivariate model, and variables with p-values less than 0.25 were included in the multivariate logistic regression analysis. To control the confounding variables and identify the related factors, multivariable logistic was fitted last. With a 95% confidence interval, the adjusted odds ratio was used to determine the strength of the association. At a p-value of 0.05, statistical significance was declared. Model goodness of fit was evaluated using the Hosmer and Lemeshow test, which yielded a p-value of 0.8786, indicating that the model was well fit.

## Ethical consideration

The Institutional Health Research Ethics Review Committee (IHRERC) of Haramaya University's College of Health and Medical Sciences granted its approval with the reference number IHRERC/230/202. The study sought permission to conduct the research from public health institutions of Deder town. After the study's purpose, data collection method, benefits, and risks were fully disclosed to the study participants, they gave their fully informed, voluntary, written, and signed consent, and the data were collected. Participants were informed that they could terminate the interviews at any moment. Confidentiality was ensured by using codes in place of personal identifiers. *For participants under 18 years of age, written informed consent was obtained from their parent/guardian.*

## Results

### Socio-demographic characteristics

A total of 418 pregnant women participated in this study, resulting in a response rate of 98%. The mean (+SD) age of the study participant was 25.8 (±3.901) years. Among participants, 409 (97.8%) were married, 175 (41.9%) had attended primary school, and 198 (47.4%) reported a family size of 1–3 "Table 1".

### Obstetrics related factors

More than half 212(50.7%) of pregnant women had previous Antenatal care follow-ups. Regarding sources of nutrition information: more than half 216(51.7%) of the respondents got their nutrition information from their close relatives, followed by 116(27.8%) friends "Table 2".

### Knowledge and Attitude of Pregnant Women about Food Taboo Practices

Food taboos were familiar to most of the study participants as 342 (81.8%) respondents had heard of food taboos. More than half 225 (53.8%) of the study participants had poor knowledge scores. The mean knowledge score was 4.64±2.129 SD with a minimum score of 2 and a maximum score of 10. Regarding attitudes, 226 (54.1%) of the respondents had unfavorable attitudes toward food taboos practice. The majority of pregnant women 279(66.7%) did not believe that vegetables cause unpleasant smells in newborns and mothers, while 272(65.1%) of pregnant women believed that honey causes abortion "Table 3".

### Eating habits of pregnant women

Among respondents, 54.1% (n = 226) and 57.4% (n = 240) reported never consuming fruits and vegetables, respectively, during the current pregnancy., while 117 (61%) and 92 (52%) consumed fruits and vegetables once per day, respectively. Regarding meal

**Table 1. Socio-demographic characteristics of pregnant women attending ANC follow-up in Deder town health facilities, Eastern Ethiopia, 2022(N=418).**

| Variables | Category | Frequency | Percent |
|---|---|---|---|
| Age groups | <=19 | 18 | 4.3 |
| | 20-24 | 144 | 34.4 |
| | 25-29 | 162 | 38.8 |
| | 30-34 | 91 | 21.8 |
| | >=35 | 3 | 0.7 |
| Ethnicity | Oromo | 346 | 82.8 |
| | Amhara | 29 | 6.9 |
| | Gurage | 23 | 5.5 |
| | Harari | 10 | 2.4 |
| | Other* | 10 | 2.4 |
| Religions of mother | Muslim | 324 | 77.5 |
| | Orthodox | 43 | 10.3 |
| | Protestant | 38 | 9.1 |
| | Waqefata | 13 | 3.1 |
| Current marital status | Married (live together) | 409 | 97.8 |
| | Divorced | 4 | 1.0 |
| | Widowed | 5 | 1.2 |
| Mother educational status | Unable to read and write | 68 | 16.3 |
| | Primary school | 175 | 41.9 |
| | Secondary school | 113 | 27.0 |
| | College and above | 62 | 14.8 |
| Occupations of mother | Housewife | 268 | 64.1 |
| | Gov't employee | 55 | 13.2 |
| | Merchant | 88 | 21.1 |
| | Other** | 7 | 1.7 |
| Husband educational level | Unable to read and write | 87 | 20.8 |
| | Primary school | 77 | 18.4 |
| | Secondary school | 144 | 34.4 |
| | College and above | 110 | 26.3 |
| Average monthly income | <=1000 | 177 | 42.3 |
| | 1001–2000 | 21 | 5.0 |
| | 2001–3000 | 120 | 28.7 |
| | 3001–4000 | 16 | 3.8 |
| | >=4000 | 84 | 20.1 |
| Residence of mother | Rural | 293 | 70.1 |
| | Urban | 125 | 29.9 |

* Somali, Silte. ** Student ***driver, daily laborer

patterns; the majority 215(51.4%) of the respondents did not eat an additional meal during the current pregnancy and the main reason for not taking additional meals was the fear of a big baby 108(50.2%) followed by the fear of weight gain 89(41.4%). Regarding the practice of skipping meals, 133(31.8%) of respondents skipped at least one meal per day "**Table 4**".

Among the 418 respondents, 211 (50.5%) had consumed coffee during the current pregnancy 155 (73.5%) had drunk less than two cups of coffee per coffee ceremony while only 56 (26.5%) drank more than two cups of coffee per coffee ceremony.

**Table 2. Obstetrics-related factors of pregnant women attending ANC follow-up in Deder town health facilities, Eastern Ethiopia, 2022(N=418).**

| Variables | Category | Frequency | Percent |
|---|---|---|---|
| Family size | 1–3 | 198 | 47.4 |
| | 4–6 | 169 | 40.4 |
| | >=7 | 51 | 12.2 |
| Number of birth(parity) | ≤ 2 | 214 | 51.2 |
| | 3-6 | 171 | 40.9 |
| | ≥ 7 | 33 | 7.9 |
| Previous ANC follow-up | Yes | 212 | 50.7 |
| | No | 206 | 49.3 |
| GA at the first ANC visit | <=12wks | 49 | 11.7 |
| | 12–27wks | 221 | 52.9 |
| | >=28wks | 148 | 35.4 |
| Sources of nutrition information | Close relatives | 216 | 51.7 |
| | Friends | 116 | 27.8 |
| | Health workers | 37 | 8.9 |
| | Others* | 49 | 11.7 |

* TV, radio, newspaper, social media

**Table 3. Knowledge, and attitudes of pregnant women attending ANC follow-up in Deder town health facilities, Eastern Ethiopia, 2022(N=418).**

| Knowledge questions | Yes N (%) | No N (%) |
|---|---|---|
| Have you heard about the taboo for pregnant women during pregnancy? | 342(81.8) | 76(18.2)* |
| Fatty/oily foods are taboo for pregnant women | 324(77.5) | 94(22.5)* |
| Milk and its products are taboo for pregnant women | 313(74.9) | 105(25.1)* |
| Bananas are taboo for pregnant women | 331(79.2) | 87(20.8)* |
| An egg is a taboo for pregnant women | 307(73.4) | 111(26.6)* |
| Potato is taboo for pregnant women | 22(5.3) | 396(94.7)* |
| Cabbage is taboo for pregnant women | 210(50.2) | 208(49.8)* |
| Honey is taboo for pregnant women | 221(52.9) | 197(47.1)* |
| Sugarcane is taboo for pregnant women | 94(22.5) | 324(77.5)* |
| Pumpkin is taboo for pregnant women | 78(18.7) | 340(81.3)* |

**Knowledge status Good** (mean score ≥5) =193(46.2%)
**Poor** (mean score <5) =225(53.8%)

| Attitude related questions | Agree N (%) | Unsure N (%) | Disagree N (%) |
|---|---|---|---|
| Vegetables coated on the fetal body cause newly born babies and mothers to bad smell | 135(32.3) | 4(1) | 279(66.7)* |
| Fatty meat and butter coated on the fetal body cause the fetus to rise down in the womb and make difficulties during delivery | 212(50.7) | 0 | 206(49.3)* |
| Read meat causes the baby too big in the womb and causes prolonged labor | 209(50) | 0 | 209(50)* |
| Fruits cause the baby too big in the womb and cause difficulties during delivery | 214(51.2) | 1(0.2) | 203(48.6)* |
| Honey cause abortion | 272(65.1) | 1(0.2) | 145(34.7)* |
| Mustered cause abortion | 135(32.3) | 4(1) | 279(66.7)* |
| spicy food causes the baby's hair to thin | 209(50) | 0 | 209(50)* |

**Attitude status Favorable** (mean score>10) =192(45.9%)
**Unfavorable** (mean score ≤10) = 226(54.1%)

* Correct answer

**Table 4. Meal patterns of pregnant women attending ANC follow-up in Deder town health facilities, Eastern Ethiopia, 2022(N=418).**

| Variables | Category | Frequency | Percentage |
|---|---|---|---|
| Regular meal eating frequency | Once | 37 | 8.9 |
| | Twice | 116 | 27.8 |
| | Three times | 216 | 51.7 |
| | ≥ Four times | 49 | 11.7 |
| Additional meal practices | Yes | 203 | 48.6 |
| | No | 215 | 51.4 |
| Number of additional meal practice | One | 127 | 62.6 |
| | Two | 58 | 28.6 |
| | Three | 18 | 8.9 |
| Reasons for not taking additional meal | Fear big baby | 108 | 50.2 |
| | Fear of weight gain | 89 | 41.4 |
| | Lack of appetite | 18 | 8.4 |
| Skipping meal practice | Yes | 133 | 31.8 |
| | No | 285 | 68.2 |
| Types of meal skipped | Breakfast | 77 | 57.9 |
| | Lunch | 26 | 19.5 |
| | Dinner | 30 | 22.6 |
| Reasons for skipping | Fear of nausea and vomiting | 38 | 28.6 |
| | Abdominal discomfort | 14 | 10.5 |
| | Heartburn | 68 | 51.1 |
| | Fasting | 13 | 9.8 |

### Food aversions and craving among pregnant women

Out of 418 respondents, 218 (52.2%) developed food aversion during the current pregnancy. The food averted were; 89(40.8%) averted cereal products, followed by eggs 77 (35.3%). The main reasons for food aversion were 121 (55.5%) due to nausea and vomiting, followed by 77 (35.3%) smell of foods. The trimester food aversion developed was 114 (52.3%) first trimester followed by, 61 (28%) third trimester "**Table 5**".

Regarding pica. among the 418 respondents, 319 (76.3%) respondents did not develop cravings for unnatural substances while only 99 (23.7%) respondents experienced the consumption of pica. Among those who experienced the consumption of pica, 78 (78.8%) consumed clay followed by 14 (14.1%) and 7 (7.1%), consumed ash and ice respectively.

### The magnitude of food taboo practices among pregnant women

Two hundred thirty-four (56%) of the respondents avoided at least one food item during the current pregnancy for various traditional beliefs and cultures with a 95% CI of 51.2–60.8%. Among those who practiced food taboos, 81 (34.6%) considered fruits as taboos, followed by 43 (18.4%) vegetables, 38 (18.4%) honey, 36(15.4%) milk products, 28 (12%) meat, and 8 (3.4%) egg. Regarding reasons for avoiding these foods; A common belief underpinning many of the taboos was difficulty during childbirth because the baby will be big 109 (46.6%), followed by 44 (18.8%) abdominal cramps for the mother and fetus, the unnecessary thing will be coated on fetal body 43 (18.4%), and causes miscarriage (15.4%).

**Table 5. Food aversions and cravings among pregnant women attending ANC follow-up in Deder town health facilities, Eastern Ethiopia, 2022(N=418).**

| Variables | Category | Frequency | Percent (%) |
|---|---|---|---|
| Food aversion | Yes | 218 | 52.2 |
| | No | 200 | 47.8 |
| Types of food averted | Cereal products (Pasta, Rice, and Spaghetti) | 89 | 40.8 |
| | Egg | 77 | 35.3 |
| | Fatty meat | 32 | 14.7 |
| | Spicy foods | 20 | 9.2 |
| Reasons for food aversion | Nausea and vomiting | 121 | 55.5 |
| | Smell of food | 77 | 35.3 |
| | Heartburn | 20 | 9.2 |
| Trimester aversion Developed | 1st trimester | 114 | 52.3 |
| | 2nd trimester | 43 | 19.7 |
| | 3rd trimester | 61 | 28 |
| Pregnant women experience food craving | Yes | 189 | 45.2 |
| | No | 229 | 54.8 |
| Types of food craved | Meat | 118 | 62.4 |
| | Cultural food | 29 | 15.3 |
| | Egg | 24 | 12.7 |
| | Injera | 12 | 6.3 |
| | Bread | 6 | 3.2 |
| Reasons for food crave | Taste of food | 84 | 44.4 |
| | Smell of food | 65 | 34.4 |
| | Color of food | 20 | 10.6 |
| | Other (like advice from others) | 20 | 10.6 |

### Factors associated with food taboo practices among pregnant women

In the multivariate analysis, pregnant women's educational status (unable to read and write), partner's educational status (primary education), no previous ANC visit, having food aversion, no additional meal practice, poor knowledge of food taboos, and unfavorable attitude toward food taboos were significantly associated with food taboo practice.

Pregnant women who were unable to read and write had three times higher odds of practicing food taboo practices (AOR=3.36, 95% CI: 1.24, 9.16) as compared to those who had attended college and above. Pregnant women who had no previous ANC follow-up had two times higher odds of practicing food taboos (AOR=2.04, 95% CI: 1.27, 3.3) compared to their counterparts.

Pregnant women with food aversion had two times higher odds of practicing food taboos (AOR=2.04, 95% CI:1.308, 3.18) compared to their counterparts. The odds of practicing food taboos were nearly two times higher among pregnant women without additional meal practices (AOR=1.8, 95% CI: 1.14, 2.76), having poor knowledge of food taboos (AOR=1.96, 95% CI: 1.24, 3.09), having unfavorable attitudes (AOR=1.91, 95% CI: 1.22, 2.99) compared to their counterparts respectively "**Table 6**".

## Discussion

Food taboos can significantly impact maternal health by restricting pregnant women from consuming certain nutritious foods. This restriction can lead to malnutrition, negatively affecting both the mother and the developing fetus [4]. This study measured the extent of practices surrounding food taboos. Furthermore, it was discovered that food taboo practices

**Table 6. Factors associated with food taboos among pregnant women attending ANC follow-up in Deder town health facilities, Eastern Ethiopia, 2022.**

| Variables | Category | Food Taboo | | COR (95% CI) | AOR (95% CI) |
|---|---|---|---|---|---|
| | | Yes No (%) | No No (%) | | |
| Educational status | Unable to read and write | 47(20.1%) | 21(11.4%) | 3.1(1.51, 6.37) * | 3.36(1.24, 9.16)** |
| | Primary school | 102(43.6% | 73(39.7%) | 1.94(1.07, 3.48)* | 1.636(0.726, 3.684) |
| | Secondary school | 59(25.2%) | 54(29.3%) | 1.513(0.810,2.83) | 0.99(0.433,2.279) |
| | College and above | 26(11.1% | 36(19.6%) | 1:00 | 1:00 |
| Partner educational status | Unable to read and write | 49(21%) | 38(20.7%) | 1.387(0.788, 2.440) | 0.904(0.399, 2.046) |
| | Primary school | 60(25.4%) | 17(9.2%) | 3.796(1.97, 7.3)* | 2.768(1.24, 6.2)** |
| | Secondary school | 72(31%) | 72(39.1%) | 1.07 (0.66, 1.77) | 0.817(0.41, 1.62) |
| | College and above | 53(22.6%) | 57(31%) | 1:00 | 1:00 |
| Residence | Rural | 144(61.5%) | 149(81%) | 2.661(1.692,4.18)* | 0.892(0.38, 2.09) |
| | Urban | 90(38.5%) | 35(19%) | 1:00 | 1:00 |
| ANC follow-up | Yes | 103(44%) | 109(59.2%) | 1:00 | 1:00 |
| | No | 131(56%) | 75(40.8%) | 1.848(1.250, 2.734) | 2.044(1.27, 3.29)** |
| Food Aversion | Yes | 140(59.8%) | 78(42.4%) | 2.024(1.37, 2.99) | 2.04(1.31, 3.20)** |
| | No | 94(40.2%) | 106(57.6%) | 1:00 | 1:00 |
| Food craving | Yes | 117(50%) | 72(39%) | 1.556(1.05, 2.3)* | 1.484(0.872, 2.524) |
| | No | 117(50%) | 112(61%) | 1:00 | 1:00 |
| Additional meal practice | Yes | 102(43.6%) | 101(54.9%) | 1:00 | 1:00 |
| | No | 132(56.4%) | 83(45.1%) | 1.575(1.068, 2.323) | 1.770(1.14, 2.76)** |
| Skipping meal | Yes | 69(29.5%) | 64(34.8%) | 1:00 | 1:00 |
| | No | 165(70.5%) | 120(65.2%) | 0.784(0.519, 1.186) | 0.996(0.570, 1.743) |
| Coffee consumption | Yes | 131(56%) | 80(43.5%) | 1.653(1.12, 2.4)* | 1.115(0.658, 1.89) |
| | No | 103(44%) | 104(56.5%) | 1:00 | 1:00 |
| Pica consumption | Yes | 61(26.1%) | 38(20.7%) | 1.355(0.854, 2.148) | 1.293(0.730, 2.290) |
| | No | 173(73.9%) | 146(79.3%) | 1:00 | 1:00 |
| Knowledge status | Poor | 150(64.1%) | 75(40.8%) | 2.595(1.74, 3.86)* | 1.959(1.24, 3.09)** |
| | Good | 84(35.9%) | 109(59.2%) | 1:00 | 1:00 |
| Attitudes status | Unfavorable | 142(60.7%) | 84(45.7%) | 1.837(1.24, 2.72)* | 1.909(1.22, 2.99)** |
| | Favorable | 92(39.3%) | 100(54.3%) | 1:00 | 1:00 |

**COR:** Crude Odds Ratio; **AOR:** Adjusted Odds Ratio *: Significant at P-value <0.25; **: P-Value < **0.05** considered as statistically significant.

were substantially correlated with pregnant women's educational status, their husbands' educational status, their ANC follow-up, their food aversion, their supplementary meal practice, their nutritional understanding, and their attitude toward nutrition.

We found that the prevalence of food taboo practices was 56% (95% confidence interval: 51.2–60.8%). This result is consistent with research conducted in the Ethiopian towns of Shashamane (49.8%) [10] Mandura (55.2%) [47], and Sendafa Bake (55.3%) [13]. This finding is higher than studies conducted in Ethiopia Mekele City (11.5%) [22] and Awebel District (27%) [12]. The possible justification could be variations in sociodemographic characteristics, research site, ANC service coverage, sample size, and sampling techniques. The results of this study were also lower than those of studies from Malay 70.2% [20] and Papua New Guinea 66% [48]. This may be due to variations in cross-cultural differences, economic status, study location, and inclusion standards. Another explanation for the high prevalence of food taboos is that women may assume food taboos can help preserve cultural traditions and practices, fostering a sense of identity and community [31]. Additionally, in some cases, food taboos

may discourage the consumption of foods that are more prone to contamination or spoilage, thus reducing the risk of foodborne illnesses some food taboos may have health benefits such as avoiding certain foods that are known allergens or that may exacerbate certain medical conditions in susceptible individuals [18]. All those conditions may lead to the high prevalence of food taboos practice.

This finding implied that food taboos are widely practiced among pregnant women across the globe. A meta-analysis study showed that pregnant women avoid certain foods due to cultural concerns about birth weight and clinical evidence of obstructed labor in Africa, Asia, and Europe [49]. Policymakers should implement community-based education campaigns, promote male partner involvement, and integrate targeted nutritional counseling into ANC visits to address food taboos. Empowering them with accurate nutritional information ensures maternal and fetal health is not compromised.

We found a strong association between education level and food taboo practices. This finding is consistent with research from Sudan, Shashamane, Mekele, and Sendafa Bake in Ethiopia [10,13,22,50]. The conceivable explanation could be that women who have no formal education are unable to acquire knowledge concerning the effects of food taboos and take into account cultural myths that promote avoiding food. Additionally, individuals could follow food taboos because they lack knowledge from formal education and cannot study useful literature.

Pregnant women whose husbands had attended primary school were more likely to observe food taboos than their counterparts. This finding is consistent with a study conducted in Sidama Ethiopia [51]. This may be because husbands with less education are unable to provide their wives with nutritional advice due to a lack of expertise. This result suggested that there is a glaring informational gap in the effects of dietary abstinence during pregnancy. Therefore, it is preferable to work on raising awareness for both women and their families.

In this study, not having ANC follow-up was significantly associated with food taboo practices. Pregnant women who have never had ANC follow-up were more likely to practice food taboos as compared with pregnant women who have had ANC follow-up. This finding is in harmony with studies conducted in Mandura and Awebel districts in Ethiopia [12,47]. The reason could be that mothers who never had ANC follow-up will miss out on nutritional counseling services and awareness-raising activities given by healthcare professionals, which will ultimately raise food taboo practices. The study's findings, which show that relatives like the mother, grandmother, mother-in-law, and friends are the pregnant women's primary sources of dietary information rather than medical facilities or other professionals, support this. This result suggested that improving ANC follow-up will enhance or improve their comprehension of the effects of food taboos.

In addition to educational factors, food aversion was strongly associated with food taboo practices. Similar trends have been observed in southwest India and Ethiopian regions like Sendafa Bake and Sidama Zone [13,31,51]. A change in olfactory and taste sensitivity, which could result in nausea, might be considered a possible factor contributing to the development of food aversion in pregnant women. It was also found that additional meal practice was also significantly associated with a pregnant woman's practice of food taboos.

Pregnant women who did not eat an extra meal during pregnancy were more likely to have food taboos than their counterparts. This finding is similar to studies conducted in Khartoum, Sudan, Sendafa Bake town, and Mandura Woreda, Ethiopia. This might be due to wrong beliefs or religious and cultural views. As identified in all studies pregnant women avoided the extra meal due to fear of gaining weight, fear of difficulty in childbirth, and lack of appetite [13,47,50].

Furthermore, the study also identified that pregnant women's attitudes were also significantly associated with food taboos. Accordingly, 54.1% of pregnant women have an unfavorable attitude toward food taboos which is in agreement with the study conducted in the Shashamane District, Ethiopia (49.8%) [10], and Pondicherry, South India (63.7%) [28]. This similarity might be due to adherence to the religious and cultural practices, educational status, and residence of study participants. Overall, addressing food taboos through education, improved ANC follow-up, and community engagement is crucial for enhancing maternal and child health outcomes.

## Strengths and limitations of the study

The strength of the study is that it assessed pregnant women's knowledge and attitudes toward food taboo practices that had not been previously evaluated by other researchers. However, since it was only a quantitative method it cannot provide an in-depth explanation of why pregnant women practice food taboos. Also, because it was a facility-based study, the results of the study cannot be generalized to all pregnant women in the study area. Furthermore, this study used both male and female interviewers and the study may be influenced by interviewer effects as women may not discuss all practices with male interviewers. Future studies could incorporate qualitative methods to explore the underlying reasons for food taboo practices among pregnant women.

## Conclusion

More than half of the pregnant women in this study were restricted from at least one food item because they believed in traditional and cultural beliefs. Pregnant women unable to read and write, pregnant women whose husbands attended primary school, had no previous ANC follow-up, had food aversion, had no additional meal practice, had poor knowledge of food taboos, and unfavorable attitudes toward food taboos were more likely to practice food taboos. The food items that were avoided during pregnancy were fruits, vegetables, honey, milk and dairy products, fatty meat, and eggs. The reasons for avoiding these food items were fear of difficulties during childbirth due to the increased size of the fetus, causing abdominal cramps to mother and fetus, attaching to the fetal body and causing an offensive odor to the mother and newborn baby, and fear of miscarriage. While some women may perceive benefits from food taboos, it is essential to address these beliefs through culturally sensitive educational programs. Implementing culturally appropriate interventions such as community workshops can empower women by educating them on the nutritional benefits of restricted foods while also dispelling harmful myths about their consumption.

## Supporting information

**S1 Data. Stata file of food taboo dataset.**
(RAR)

## Acknowledgments

We would like to express our sincere gratitude to Haramaya University for supporting us in writing this work, as well as to the data collectors and supervisors.

## Author contributions

**Conceptualization:** Abdi Tofik, Tesfaye Gobena, Addis Eyeberu, Adera Debella, Berhe Gebremichael, Mulugeta Gamachu, Alemayehu Deressa, Galana Mamo Ayana, Hamdi Fekredin Zakaria, Usmael Jibro, Ibsa Mussa.

**Data curation:** Abdi Tofik, Tesfaye Gobena, Adera Debella, Alemayehu Deressa, Abdi Birhanu, Hamdi Fekredin Zakaria.

**Formal analysis:** Tesfaye Gobena, Addis Eyeberu, Berhe Gebremichael, Galana Mamo Ayana, Abdi Birhanu, Hamdi Fekredin Zakaria, Usmael Jibro, Ibsa Mussa.

**Investigation:** Abdi Tofik, Mulugeta Gamachu, Galana Mamo Ayana, Ibsa Mussa.

**Methodology:** Galana Mamo Ayana.

**Software:** Abdi Tofik, Berhe Gebremichael, Alemayehu Deressa.

**Supervision:** Tesfaye Gobena, Berhe Gebremichael, Abdi Birhanu, Ibsa Mussa.

**Validation:** Berhe Gebremichael.

**Writing – original draft:** Abdi Tofik, Addis Eyeberu.

**Writing – review & editing:** Addis Eyeberu, Berhe Gebremichael, Usmael Jibro.

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
