## [Decision Letter · Decision Letter 0]

7 Jan 2025

PONE-D-24-22278Food taboo practices among pregnant women in Deder town, Eastern Ethiopia, 2024PLOS ONE

Dear Dr. Jibro,

Thank you for submitting your manuscript to PLOS ONE. After careful consideration, we feel that it has merit but does not fully meet PLOS ONE’s publication criteria as it currently stands. Therefore, we invite you to submit a revised version of the manuscript that addresses the points raised during the review process.

We look forward to receiving your revised manuscript.

Kind regards,

Sandra Boatemaa Kushitor, Ph.D.

Academic Editor

PLOS ONE

Journal Requirements:

2.  In the online submission form, you indicated that [Data available on request from the corresponding author].

Additional Editor Comments:

Dear Authors,

Kindly address the comments I have raised and those of the reviewer.

Thank you.

Reviewers' comments:

Reviewer's Responses to Questions

**Comments to the Author**

1. Is the manuscript technically sound, and do the data support the conclusions?

Reviewer #1: Yes

2. Has the statistical analysis been performed appropriately and rigorously? 

Reviewer #1: Yes

3. Have the authors made all data underlying the findings in their manuscript fully available?

Reviewer #1: Yes

4. Is the manuscript presented in an intelligible fashion and written in standard English?

Reviewer #1: Yes

5. Review Comments to the Author

Reviewer #1: Below, I summarize my evaluation:

1. Technical Soundness: The study is technically sound, with well-conducted statistical analysis. The data provided adequately supports the conclusions drawn.

2. Statistical Analysis: The statistical analyses are rigorous and appropriate for the study design. The authors have clearly described their methods and adhered to best practices in reporting and interpretation.

3. Data Availability: The authors have complied with the PLOS ONE data-sharing policy, making all data underlying the findings available without restriction. The Data Availability Statement is clear and detailed.

4. Manuscript Presentation: The manuscript is well organized, written in clear and standard English, with minor typographical or grammatical errors. Minor improvements to phrasing or formatting could further enhance clarity but are not critical.

Additional Comments

The manuscript provides valuable insights into food taboos and is a meaningful contribution to the field. I have no major concerns regarding the research ethics, data availability, or publication ethics. Minor revisions to address points in my review of this manuscript could strengthen the presentation.

Overall, I recommend this manuscript for publication pending minor revisions.

Below is the full review:

Manuscript Number: PONE-D-24-22278

Manuscript Title: Food taboo practices among pregnant women in Deder town, Eastern Ethiopia, 2024

Abstract

Background

The background effectively sets the context for the study but could be more specific. Consider briefly mentioning why food taboos are significant and how they impact maternal and foetal health.

Grammatical Corrections:

• Line 25-27: Consider rephrasing to "Maternal nutrition during pregnancy is influenced by food taboo practices, which vary across cultural contexts. Understanding these practices in Eastern Ethiopia is crucial for designing culturally appropriate interventions."

Method

You might want to include more details about the sampling method (e.g., random sampling, convenience sampling) to give readers insight into how participants were selected.

The phrase "56% ((95% CI: 51.2–60.8%)" has an extra parenthesis. Correct this for precision

Results

Ensure consistency in presenting confidence intervals (CIs). For example, use either "AOR=2.04" or "AOR: 2.04" throughout for uniformity.

• Consider clarifying what "ANC" stands for when first mentioned (Antenatal Care) to ensure all readers understand the acronym.

Conclusion

While the recommendation for "nutrition education and awareness creation" is valid, the conclusion should emphasize the need for culturally tailored strategies, given the study's context.

• Suggested revision:

"More than half of pregnant women practiced food taboos, indicating a significant public health concern. Culturally sensitive nutrition education and awareness programs at health facilities are necessary to address these practices and improve maternal nutrition outcomes."

Introduction

1. Opening Sentences: The introduction begins with a definition of food taboos, which is effective. However, it could benefit from a more engaging opening that highlights the significance of the topic. Consider starting with a statement about the importance of maternal nutrition or the prevalence of food taboos globally. Also, consider rephrasing the first sentence (Line 48) to: "Food taboos refer to dietary restrictions influenced by religious, cultural, or health beliefs.

2. Contextualization of Studies: When referencing studies from other countries (e.g., Malaysia, South Africa), briefly explain how these findings relate to the Ethiopian context. This will strengthen the rationale for your study by showing how it fits into a broader framework.

3. Cultural Context: The introduction mentions that Eastern Ethiopia's cultural context differs from other regions but does not specify how or why this is significant. Providing specific cultural beliefs or practices related to food taboos in this region could enhance understanding.

4. Research Gap: While you mention that food avoidance during pregnancy is an unresolved health concern, explicitly stating what previous studies have found lacking in Eastern Ethiopia would clarify the research gap your study aims to fill.

5. Minor Technical Suggestions:

Line 58: Add a full-stop after "learning"

Line 83: The statement about food taboos being potentially adaptive seems abrupt and could use more context

Ensure consistent formatting of citations (some have spaces, some don't)

6. Clarity and Flow:

Global Prevalence and Context (Lines 59–66):

The presentation of prevalence rates across countries lacks a logical flow. Transitioning from global to regional (Africa) and then to Ethiopia would improve readability.

Suggested Structure:

Global: "Globally, food taboo practices are prevalent, with rates reported at 70.2% in Malaysia [16], 37% in South Africa [13], and 66% in Nigeria [17]."

Regional (Ethiopia): "In Ethiopia, reported prevalence ranges from 11.5% to 55.3% [7, 9, 10, 18], with a systematic review estimating an average of 34.22% [11]."

7. Precision in Language:

"Widespread food taboo practices around the world" (Line 60): Avoid vague phrases like "widespread" in technical writing.

Suggested Revision: "Food taboo practices have been reported globally, with varying prevalence across regions."

"Sizable portion of pregnant women" (Line 71): Replace with specific or technical terms.

Suggested Revision: "A significant proportion of pregnant women..."

8. Evidence and Citations:

Line 75: "Women have strong beliefs in their culture" is vague and not fully supported. Provide a clearer, evidence-based statement.

Suggested Revision: "Cultural traditions strongly influence dietary practices during pregnancy, as women often inherit food taboos through familial and societal transmission [7, 13]."

Line 86–88: The interventions listed (e.g., nutritional counseling, micronutrient supplementation) are well-referenced but could briefly mention their implementation gaps to tie into the study rationale.

"Despite interventions such as nutritional counseling and micronutrient supplementation [28–33], gaps remain in addressing culturally driven food taboos."

9. Study Rationale and Objective:

The study's rationale is clear but somewhat repetitive. Condense the justification and emphasize the research gap more succinctly.

Suggested Revision:

"While Ethiopia has made strides in improving maternal nutrition, food avoidance during pregnancy remains a significant concern, particularly in Eastern Ethiopia, where cultural contexts differ. Understanding these practices is crucial to designing targeted, culturally sensitive interventions. This study aims to assess the extent and determinants of food taboos among pregnant women in Eastern Ethiopia."

MATERIALS AND METHODS

1. Study Design and Area

Geographical Context: Consider including more information about the socio-economic context or health infrastructure that might influence maternal health practices.

2. Populations and Criteria

• Consider rephrasing to: "The study included all pregnant women attending ANC follow-ups at Deder Hospital and health centers, except those unable to respond during the data collection period."

3. Sample Size Determination and Sampling Procedure

Sample Size Calculation: Briefly explain the significance of a 10% non-response rate which would also clarify its importance.

The description of the sample size calculation is clear but could benefit from a formula citation or equation.

Also, the sampling procedure explanation is slightly wordy.

Suggestion:

"Systematic random sampling was used to select participants using ANC identification numbers, with the first sample chosen randomly between the first and second attendees."

4. Data Collection Procedure and Quality Control

• The phrase "Seven different health professions gathered the data" is unclear. Consider clarifying the roles, eg: "Data were collected by seven trained health professionals, including [specify roles if possible]."

• The phrase "Data collection were started on February 01, 2022 and ends on February 30, 2022" contains grammatical errors.

• Briefly explain the purpose of Cronbach's alpha for readers unfamiliar with it:

"The reliability of the questionnaire was assessed using Cronbach's alpha, yielding a value of 0.83, indicating good internal consistency."

5. Model Evaluation: Mentioning that Hosmer and Lemeshow tests were used for goodness-of-fit evaluation is important. Consider briefly explaining what this entails for readers who may not be familiar with this statistical method.

6. Ethical Considerations

The ethical considerations are adequately addressed. It might be useful to mention how confidentiality was maintained during data collection and whether participants could withdraw from the study at any time without repercussions.

7. Minor Technical Corrections:

Line 142: "Data collection started" instead of "Data collection were started"

"Ends" should be "ended" on the same line

Some citation formatting could be standardized

RESULTS

1. Minor Formatting Issues:

• Typographical error in line 208: Extra percentage sign

• Ensure consistent formatting of statistical presentations

2. I suggest authors alter this sentence: "Of the total study participants, 409(97.8%), 175(41.9%), and 198(47.4%) women who were married attended primary school and had a family size of 1-3 respectively (Table 1)" to “Among participants, 409 (97.8%) were married, 175 (41.9%) had attended primary school, and 198 (47.4%) reported a family size of 1–3 (Table 1)."

3. Authors should re-consider this sentence: "Of the 418 respondents, 226 (54.1%) and 240 (57.4%) never consumed fruits and vegetables respectively during the current pregnancy..." to "Among respondents, 54.1% (n = 226) and 57.4% (n = 240) reported never consuming fruits and vegetables, respectively, during the current pregnancy."

DISCUSSION

1. Introduction to the Discussion: The opening sentences effectively summarize the main findings regarding food taboo practices and their associations with various factors. However, consider starting with a broader statement about the significance of food taboos in maternal health to engage readers more effectively.

2. Minor Writing Improvements:

Line 276-279: Some grammatical refinements needed

Ensure consistent formatting of citations

3. Lines 271–273:

This sentence lacks actionable detail "Policymakers must prioritize education and awareness campaigns to address prevalent food taboos among pregnant women." Consider changing it to: "Policymakers should implement community-based education campaigns, promote male partner involvement, and integrate targeted nutritional counseling into ANC visits to address food taboos."

4. Consider rephrasing this sentence "We found a strong correlation between food aversion and food taboo practices. Studies conducted in southwest India, Sendafa Bake town, Ethiopia, and Sidama Zone, Ethiopia, confirm this conclusion (10, 27, 45)" to "In addition to educational factors, food aversion was strongly associated with food taboo practices. Similar trends have been observed in southwest India and Ethiopian regions like Sendafa Bake and Sidama Zone (10, 27, 45)." In lines 296-298

5. The concluding sentence could benefit from a strong conclusion statement, eg: Overall, addressing food taboos through education, improved ANC follow-up, and community engagement is crucial for enhancing maternal and child health outcomes."

STRENGTHS AND LIMITATIONS

The limitation regarding the use of only quantitative methods is well-stated. To strengthen this point, consider suggesting how qualitative methods (e.g., interviews or focus groups) could complement your findings. For instance: "Future studies could incorporate qualitative methods to explore the underlying reasons for food taboo practices among pregnant women."

CONCLUSION

• Cultural Context: The mention of perceived advantages to food taboos is important. It would strengthen your conclusion to briefly explain how these perceived advantages can be addressed in interventions. For example, "While some women may perceive benefits from food taboos, it is essential to address these beliefs through culturally sensitive educational programs."

• Nutritional Counseling: The recommendation for culturally appropriate interventions and nutritional counseling during ANC visits is a strong point. You might want to specify what these interventions could entail. For example, "Culturally appropriate interventions might include community workshops that educate women about the nutritional value of restricted foods and debunk myths surrounding their consumption."

Other: Editing to resolve typo related issues

6. PLOS authors have the option to publish the peer review history of their article (what does this mean? ). If published, this will include your full peer review and any attached files.

**Do you want your identity to be public for this peer review?** For information about this choice, including consent withdrawal, please see our Privacy Policy .

Reviewer #1: **Yes: ** Marian Yenupini Kombat

---

## [Author Response · Author response to Decision Letter 1]

6 Feb 2025

Point by point response to editor and reviewers

Title: Food taboo practices among pregnant women in Deder town, Eastern Ethiopia, 2024

Manuscript ID: PONE-D-24-22278

From: Authors

To: The editor in chief, PLOS ONE

Version: I

Data: 25/1/2025

Subject: Revision of the manuscript

We appreciate the reviewers' detailed and comprehensive comments. We found the comments to be very helpful, and we appreciate the time and thought that each person put into their constructive comments. We are well aware of the time, commitment required to provide good reviews and applaud the reviewers for their efforts. We thoroughly revised the paper and responded in detail to the reviewers' questions and comments.The point-by-point description of the changes is provided below.

For editors

Comment 1: Please ensure that your manuscript meets PLOS ONE's style requirements, including those for file naming.

Response 1: thank you for your comment. Now the revised version is edited based on PLOS ONE's style requirements.

2. In the online submission form, you indicated that [Data available on request from the corresponding author]. All PLOS journals now require all data underlying the findings described in their manuscript to be freely available to other researchers, either 1. In a public repository, 2. Within the manuscript itself, or 3. Uploaded as supplementary information. This policy applies to all data except where public deposition would breach compliance with the protocol approved by your research ethics board. If your data cannot be made publicly available for ethical or legal reasons (e.g., public availability would compromise patient privacy), please explain your reasons on resubmission and your exemption request will be escalated for approval.

Response 2: Thank you very much. We have now added more information about about data availability.

Response 3: Thank you for your comment. It is corrected and incorporated into the revised document. Ethics statement was omitted from declaration section.

Response 4: We are intrigued by your comment, and we accept it; it has been corrected and incorporated into the revised document.

For reviewer 1

Abstract

Comment 1. The background effectively sets the context for the study but could be more specific. Consider briefly mentioning why food taboos are significant and how they impact maternal and foetal health.

Response 1: Thank you so much for invaluable comments. It is corrected in the revised manuscript based on reviewer’s suggestion.

Comment 2. Grammatical Corrections:

• Line 25-27: Consider rephrasing to "Maternal nutrition during pregnancy is influenced by food taboo practices, which vary across cultural contexts. Understanding these practices in Eastern Ethiopia is crucial for designing culturally appropriate interventions.".

Response 2: Thank you so much for productive comments. Manuscript were revised manuscript based on reviewer’s suggestion.

Comment 3. You might want to include more details about the sampling method (e.g., random sampling, convenience sampling) to give readers insight into how participants were selected.

Response 3: Thank you for your insightful comment. We modified the manuscript as reviewer’s comment in the revised manuscript.

Comment 4. The phrase "56% ((95% CI: 51.2–60.8%)" has an extra parenthesis. Correct this for precision.

Response 4: We are intrigued by your comment, and we accept it; it has been corrected and incorporated into the revised document.

Comment 5. Ensure consistency in presenting confidence intervals (CIs). For example, use either "AOR=2.04" or "AOR: 2.04" throughout for uniformity.

Response 5: Thank you for your valuable comment. Now we have updated and incorporated in the revised manuscript.

Comment 6. Consider clarifying what "ANC" stands for when first mentioned (Antenatal Care) to ensure all readers understand the acronym.

Response 6: Thank you for your comment and suggestion. It is corrected and incorporated in the revised manuscript.

Conclusion

Comment 7. While the recommendation for "nutrition education and awareness creation" is valid, the conclusion should emphasize the need for culturally tailored strategies, given the study's context.

• Suggested revision:

"More than half of pregnant women practiced food taboos, indicating a significant public health concern. Culturally sensitive nutrition education and awareness programs at health facilities are necessary to address these practices and improve maternal nutrition outcomes."

Response 7: Thank you for your insightful and wonderful comment. We accept your comment and we incorporate it into the revised manuscript.

Introduction

Comment 1. Opening Sentences: The introduction begins with a definition of food taboos, which is effective. However, it could benefit from a more engaging opening that highlights the significance of the topic. Consider starting with a statement about the importance of maternal nutrition or the prevalence of food taboos globally. Also, consider rephrasing the first sentence (Line 48) to: "Food taboos refer to dietary restrictions influenced by religious, cultural, or health beliefs..

Response 1: Thank you for your comment. We accept it and corrected it in the revised manuscript.

Comment 2. Contextualization of Studies: When referencing studies from other countries (e.g., Malaysia, South Africa), briefly explain how these findings relate to the Ethiopian context. This will strengthen the rationale for your study by showing how it fits into a broader framework.

Response 2: Thank you for your specific comment. We accept it and corrected it in the revised manuscript.

Comment 3. Cultural Context: The introduction mentions that Eastern Ethiopia's cultural context differs from other regions but does not specify how or why this is significant. Providing specific cultural beliefs or practices related to food taboos in this region could enhance understanding

Response 3. Thank you for your insightful comment. We revised the introduction section based on your recommendations.

Comment 4. Research Gap: While you mention that food avoidance during pregnancy is an unresolved health concern, explicitly stating what previous studies have found lacking in Eastern Ethiopia would clarify the research gap your study aims to fill.

Response 4. Thank you for your productive comment. Now we have modified manuscript based on reviewers suggestion.

Comment 5. Minor Technical Suggestions:

Line 58: Add a full-stop after "learning"

Line 83: The statement about food taboos being potentially adaptive seems abrupt and could use more context

Ensure consistent formatting of citations (some have spaces, some don't)

Response 5. Thank you for your wonderful and eye-catching comments. We accept it and revise accordingly.

Comment 6. Clarity and Flow:

Global Prevalence and Context (Lines 59–66):

The presentation of prevalence rates across countries lacks a logical flow. Transitioning from global to regional (Africa) and then to Ethiopia would improve readability.

Suggested Structure:

Global: "Globally, food taboo practices are prevalent, with rates reported at 70.2% in Malaysia [16], 37% in South Africa [13], and 66% in Nigeria [17]."

Regional (Ethiopia): "In Ethiopia, reported prevalence ranges from 11.5% to 55.3% [7, 9, 10, 18], with a systematic review estimating an average of 34.22% [11].".

Response 6: Thanks for your eagle eyes. we are already changed it as suggested.

Comment 7. Precision in Language:

"Widespread food taboo practices around the world" (Line 60): Avoid vague phrases like "widespread" in technical writing.

Suggested Revision: "Food taboo practices have been reported globally, with varying prevalence across regions."

"Sizable portion of pregnant women" (Line 71): Replace with specific or technical terms.

Suggested Revision: "A significant proportion of pregnant women..."

Response 7: thank you for your insightful suggestion. Now the revised version is modified and corrected.

Comment 8. Evidence and Citations:

Line 75: "Women have strong beliefs in their culture" is vague and not fully supported. Provide a clearer, evidence-based statement.

Suggested Revision: "Cultural traditions strongly influence dietary practices during pregnancy, as women often inherit food taboos through familial and societal transmission [7, 13]."

Line 86–88: The interventions listed (e.g., nutritional counseling, micronutrient supplementation) are well-referenced but could briefly mention their implementation gaps to tie into the study rationale.

"Despite interventions such as nutritional counseling and micronutrient supplementation [28–33], gaps remain in addressing culturally driven food taboos."

Response 8: Thank you so much for fruitful comments. We have corrected and incorporated to revised manuscript based on reviewer’s suggestion.

Comment 9. Study Rationale and Objective:

The study's rationale is clear but somewhat repetitive. Condense the justification and emphasize the research gap more succinctly.

Suggested Revision:

"While Ethiopia has made strides in improving maternal nutrition, food avoidance during pregnancy remains a significant concern, particularly in Eastern Ethiopia, where cultural contexts differ. Understanding these practices is crucial to designing targeted, culturally sensitive interventions. This study aims to assess the extent and determinants of food taboos among pregnant women in Eastern Ethiopia."

Response 9: Thank you for your comment. It is corrected and incorporated in the revised manuscript.

MATERIALS AND METHODS

Comment 1. Study Design and Area

Geographical Context: Consider including more information about the socio-economic context or health infrastructure that might influence maternal health practices.

Response 1: Thank you for your insightful comment. We have modified the manuscript as per reviewer’s comment in the revised manuscript

Comment 2. Populations and Criteria

• Consider rephrasing to: "The study included all pregnant women attending ANC follow-ups at Deder Hospital and health centers, except those unable to respond during the data collection period."

Response 2: Thank you for your constructive comments. We have updated the manuscript as per reviewer’s comment in the revised manuscript.

Comment 3: Sample Size Determination and Sampling Procedure

Sample Size Calculation: Briefly explain the significance of a 10% non-response rate which would also clarify its importance.

The description of the sample size calculation is clear but could benefit from a formula citation or equation. Also, the sampling procedure explanation is slightly wordy.

Suggestion:

"Systematic random sampling was used to select participants using ANC identification numbers, with the first sample chosen randomly between the first and second attendees."

Response 3: Thank you for your insightful comment. We have modified the manuscript as per reviewer’s comment in the revised manuscript.

Comment 4: Data Collection Procedure and Quality Control

• The phrase "Seven different health professions gathered the data" is unclear. Consider clarifying the roles, eg: "Data were collected by seven trained health professionals, including [specify roles if possible]."

• The phrase "Data collection were started on February 01, 2022 and ends on February 30, 2022" contains grammatical errors.

• Briefly explain the purpose of Cronbach's alpha for readers unfamiliar with it:

"The reliability of the questionnaire was assessed using Cronbach's alpha, yielding a value of 0.83, indicating good internal consistency."

Response 4: Thank you for your constructive comment. We have modified the manuscript as per reviewer’s comment in the revised manuscript.

Comment 5: Model Evaluation: Mentioning that Hosmer and Lemeshow tests were used for goodness-of-fit evaluation is important. Consider briefly explaining what this entails for readers who may not be familiar with this statistical method.

Response 5: Thank you for your insightful comment. We have incorporated the changes to revised manuscript.

Comment 6: Ethical Considerations

The ethical considerations are adequately addressed. It might be useful to mention how confidentiality was maintained during data collection and whether participants could withdraw from the study at any time without repercussions.

Response 6: Thank you for your helpful comment. We have modified the manuscript as per reviewer’s comment in the revised manuscript.

Comment 7. Minor Technical Corrections:

Line 142: "Data collection started" instead of "Data collection were started"

"Ends" should be "ended" on the same line

Some citation formatting could be standardized

Response 7: Thank you so much for invaluable comments. It is corrected in the revised manuscript based on reviewer’s suggestion.

RESULTS

Comment 1: Minor Formatting Issues:

• Typographical error in line 208: Extra percentage sign

• Ensure consistent formatting of statistical presentations

Response 1: Dear reviewer, we gratefully received your constructive suggestion. It is corrected and incorporated in the revised manuscript.

Comment 2: I suggest authors alter this sentence: "Of the total study participants, 409(97.8%), 175(41.9%), and 198(47.4%) women who were married attended primary school and had a family size of 1-3 respectively (Table 1)" to “Among participants, 409 (97.8%) were married, 175 (41.9%) had attended primary school, and 198 (47.4%) reported a family size of 1–3 (Table 1)."

Response 2: Thank you for your insightful comment. We have modified the manuscript as per reviewer’s comment in the revised manuscript.

Comment 3: Authors should re-consider this sentence: "Of the 418 respondents, 226 (54.1%) and 240 (57.4%) never consumed fruits and vegetables respectively during the current pregnancy..." to "Among respondents, 54.1% (n = 226) and 57.4% (n = 240) reported never consuming fruits and vegetables, respectively, during the current pregnancy."

Response 3: Thank you for your comment. It is corrected and incorporated in the revised manuscript.

Discussion

Comment 1: Introduction to the Discussion: The opening sentences effectively summarize the main findings regarding food taboo practices and their associations with various factors. However, consider starting with a broader statement about the significance of food taboos in maternal health to engage readers more effectively.

Response 1: Thank you for your constructive comment. We have made changes and incorporated in the revised manuscript.

Comment 2: Minor Writing Improvements:

Line 276-279: Some grammatical refinements needed

Ensure consistent formatting of citations.

Response 2: thank you for your constructive comment. We have corrected and incorporated in revised version of manuscript.

Comment 3: Lines 271–273:

This sentence lacks actionable detail "Policymakers must prioritize education and awareness campaigns to address prevalent food taboos among pregnant women." Consider changing it to: "Policymakers should implement community-based education campaigns, promote male partner involvement, and integrate targeted nutritional counseling into ANC visits to address food taboos."

Response 3: Thank you for your comment and suggestion. It is corrected and incorporated in the revised manuscript.

Comment 4: Consider rephrasing this sentence "We found a strong correlation between food aversion and food taboo practices. Studies conducted in southwest India, Sendafa Bake town, Ethiopia, and Sidama Z

---

## [Editor Report · Decision Letter 1]

12 Mar 2025

Food taboo practices among pregnant women in Deder town, Eastern Ethiopia, 2024

PONE-D-24-22278R1

Dear Dr. Tofik,

We’re pleased to inform you that your manuscript has been judged scientifically suitable for publication and will be formally accepted for publication once it meets all outstanding technical requirements.

Kind regards,

Sandra Boatemaa Kushitor, Ph.D.

Academic Editor

PLOS ONE

Additional Editor Comments (optional):

Dear Authors,

Congratulations!!!
---

## [Editor Report · Acceptance letter]

PONE-D-24-22278R1

PLOS ONE

Dear Dr. Jibro,

I'm pleased to inform you that your manuscript has been deemed suitable for publication in PLOS ONE. Congratulations! Your manuscript is now being handed over to our production team.

Kind regards,

on behalf of

Dr. Sandra Boatemaa Kushitor

Academic Editor

PLOS ONE